# Rethinking Conventional Wisdom in Machine Learning: From Generalization to Scaling

## Abstract

The remarkable success of large language pretraining and the discovery of the empirical scaling laws signify a paradigm shift in machine learning. Notably, the primary objective has evolved from minimizing generalization error to reducing approximation error, and the most effective strategy has transitioned from regularization (in a broad sense) to scaling up models. This raises a critical question:

*Do the established principles that proved successful in the generalization-centric era remain valid in this new era of scaling?*

This paper examines several influential regularization-based principles that may no longer hold true in the scaling-centric, large language model (LLM) era. These principles include explicit L2 regularization and implicit regularization through small batch sizes and large learning rates. Additionally, we identify a new phenomenon termed "scaling law crossover," where two scaling curves intersect at a certain scale, implying that methods effective at smaller scales may not generalize to larger ones. Together, these observations highlight two fundamental questions within this new paradigm:

- **Guiding Principles for Scaling:** If regularization is no longer the primary guiding principle for model design, what new principles are emerging to guide scaling?
- **Model Comparison at Scale:** How to reliably and effectively compare models at the scale where only a single experiment is feasible?

## 1 Introduction

The advent of large language pretraining (Devlin et al., 2018; Radford et al., 2019; Raffel et al., 2020; Brown et al., 2020) and the emergence of scaling laws (Kaplan et al., 2020; Hoffmann et al., 2022; Achiam et al., 2023; Hestness et al., 2017) have led to a new paradigm within machine learning. This shift redirects both the primary objective and primary approach from optimizing *generalization* (Zhang et al., 2021) on a fixed small dataset using *regularization* to reducing *approximation* error on a huge text corpus using *scale.*

More precisely, in the previous paradigm, we have far more compute than needed to interpolate the training set. To improve the performance of the model on unseen data, we need to reduce the degree of overfitting via all kinds of regularization, including inductive biases, explicit regularization (e.g., L2 Regularization), implicit regularization (e.g., large learning rate, small batch size), among many others. By contrast, in the new paradigm, we are compute constraint, i.e., we have far more data and our compute is not enough to fit such data perfectly. Existing work shows, empirically, the better the model memorizes the data (smaller approximation error), the more powerful the (foundational) model is. As such, the most effective approach to improve performance (of downstream tasks) is to throw more compute to consume the data, aka scaling up our models (bench authors, 2023).

To distinguish these two paradigms, we refer to the former as "generalization-centric paradigm" and the latter as "scaling-centric paradigm." We summarize several key differences below in Fig. 1.

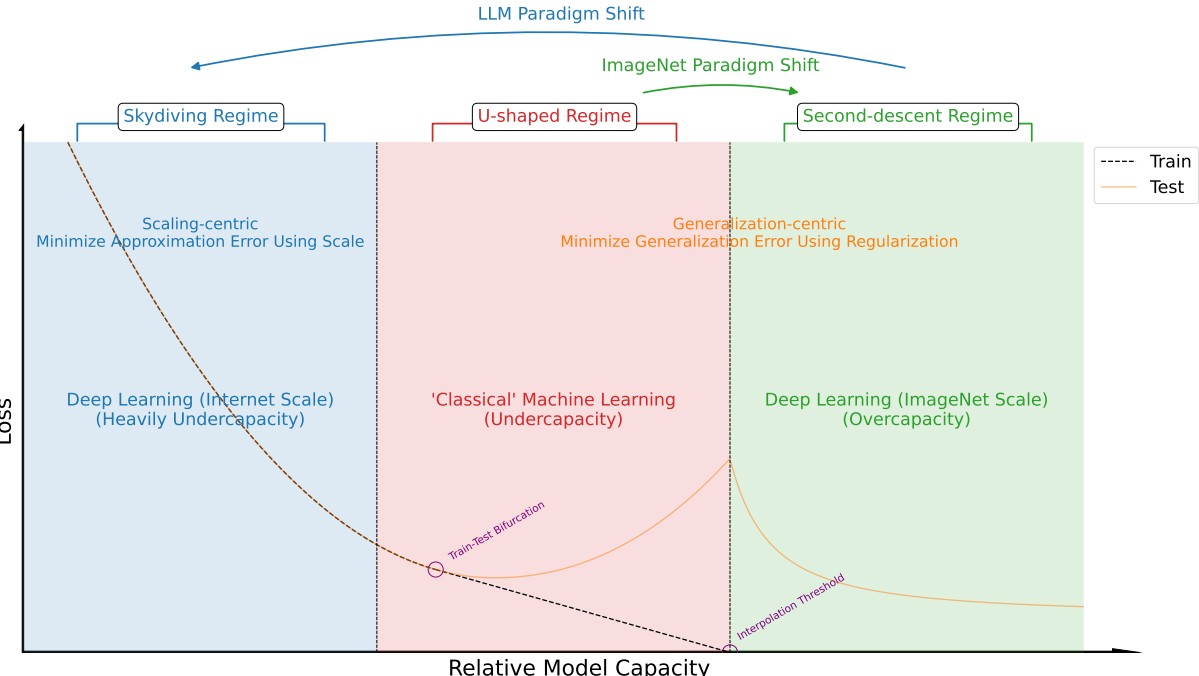

Figure 1: A proposal to reconcile "Classical" Machine Learning (U-shape), ImageNet-scale Deep Learning (Second-descent) and Internet-scale Deep Learning (Skydiving).

1. **Objective**: minimizing generalizaion error vs. minimizing approximation error.

2. **Approach**: regularization vs. scaling.

Given these fundamental differences, blindly applying best practices from the generalization-centric paradigm to the scaling-centric paradigm may be detrimental. This note revisits several influential ideas that likely originated from the need for regularization aimed at reducing overfitting:

- Larger earning rates near the maximum stable learning rate (Lewkowycz et al., 2020; Li et al., 2019) usually generalize better.

- Small batch sizes (Keskar et al., 2016; Smith & Le, 2017) usually generalize better.

Our empirical evidence suggests that several pieces of conventional wisdom relating to regularization in machine learning may not hold true in the scaling-centric paradigm. This raises a crucial question:

*"If regularization (reducing overfitting) is no longer the primary guiding principle for designing ML models, what new principles are emerging to guide scaling?"*

Another key distinction between the new paradigm and the old one is the immense scale involved (OpenAI et al., 2023; Anil et al., 2023; Zhang et al., 2024), which presents significant challenges both theoretically and practically. Notably, we observe a phenomenon termed "scaling law crossover": techniques that enhance performance at smaller scales may not translate effectively to larger ones, i.e., these techniques "overfit" to small scales. We illustrate this phenomenon with three examples. This raises a fundamental question in machine learning:

*"Given the potential for scaling law crossovers, how can we effectively compare models at a scale where only a single experiment is feasible?"*

Consequently, this new paradigm necessitates the development of novel ideas and mindsets to understand and improve scaling. We hope this work stimulates more discussion and out-of-the-box thinking on these critical challenges, moving the field forward.

## 2 Background: Two Paradigms in Machine Learning

The central goal in machine learning is to learn a function capable of making predictions on unseen data by understanding the underlying structure of the data. Formally, given a training set $\mathcal{T} = \{(\mathbf{x}_i, y_i)_{1 \leq i \leq |\mathcal{T}|}\}$ drawn from some distribution $(\mathbf{x}, y) \sim \mathcal{D}$, we aim to learn a function, parameterized by $\theta \in \Omega$, that minimizes the test loss[1] on unseen data:

$$\mathcal{L}(f_\theta; \mathcal{D}) = \mathbb{E}_{(\mathbf{x},y)\sim\mathcal{D}}[\ell(f_\theta(\mathbf{x}), y)] \tag{1}$$

In practice, we learn $f_\theta$ by minimizing the training loss (the average loss on the training set):

$$\mathcal{L}(f_\theta; \mathcal{T}) := \frac{1}{|\mathcal{T}|} \sum_{(\mathbf{x},y)\in\mathcal{T}} \ell(f_\theta(\mathbf{x}), y) \tag{2}$$

This approach is known as empirical risk minimization (ERM). The test error can be decomposed into the sum of generalization gap and training error (approximation error)

$$\underbrace{\mathcal{L}(f_\theta; \mathcal{D})}_{\text{Test Error}} = \underbrace{(\mathcal{L}(f_\theta; \mathcal{D}) - \mathcal{L}(f_\theta; \mathcal{T}))}_{\text{Generalization Gap}} + \underbrace{\mathcal{L}(f_\theta; \mathcal{T})}_{\text{Approximation Error}} \tag{3}$$

We discuss two paradigms in machine learning, distinguished by the relative and absolute scales[2] of the data and model:

- **Generalization-centric paradigm:**

  Data scale is relatively small. This paradigm further divides into two sub-paradigms:

  - **Classical bias-variance trade-off (U-shaped) regime:** Model capacity is intentionally constrained below the interpolation threshold (red dot • in Fig. 2). In this regime, both *Generalization Gap* and *Approximation Error* are non-negligible.
  - **Modern over-parameterized (second-descent) regime:** Model scale significantly surpasses data scale (green dot • in Fig. 2). In this regime, *Approximation Error* is negligible.

- **Scaling-centric paradigm:** Large data and model scales, with data scale exceeding model scale (blue dot • in Fig. 2). In this regime, the *Generalization Gap* is negligible.

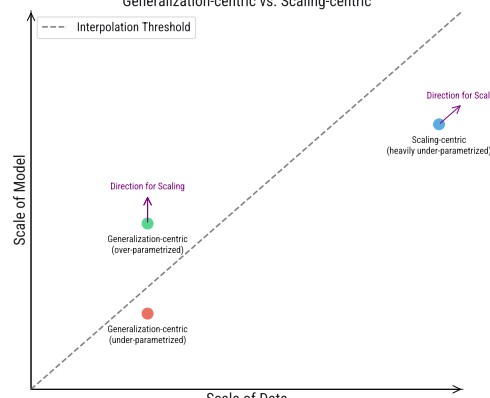

Figure 2: Generalization vs Scaling Paradigms.

---

[1]For simplicity, we assume the irreducible loss is zero.

[2]We use the term "scale" broadly in this context to encompass both the capacity of the model and the complexity of the data.

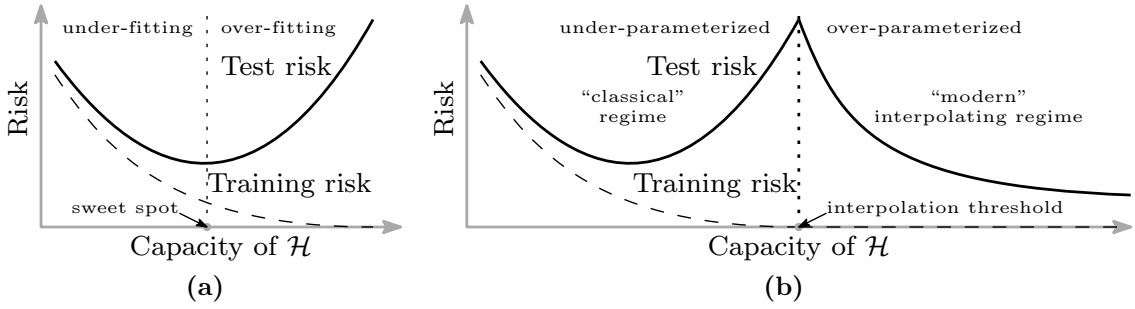

Figure 3: **U-Shaped (a) and double-descent curves (b). Figure is from Belkin et al. (2019)**

## 2.1 The Bias-Variance Trade-off and the U-shape Regime

In this classical setting, the complexity of the training set $\mathcal{T}$ is typically smaller than the richness of the function class $\mathcal{H} = \{f_\theta : \theta \in \Omega\}$ and the absolute scales of the data and models are small. Conventional wisdom in machine learning suggests that one needs to carefully control the complexity of the function space $\mathcal{H}$ (Belkin et al., 2019) to balance *Generalization Gap* and *Approximation Error*:

1. If $\mathcal{H}$ is too small (underfitting), all functions in $\mathcal{H}$ will have high bias, i.e. high *Approximation Error*. This leads to a large training error, and thus a large test error.

2. If $\mathcal{H}$ is too large (overfitting), the learned function may overfit the training data, leading to high variance. This results in a small training error but a large *Generalization Gap* (the difference between test and training errors), and thus a large test error.

This observation is known as the bias-variance trade-off, a fundamental result in machine learning (Hastie et al., 2009). It traditionally suggests a U-shaped error curve (Figure 3 (a)), with optimal test error achieved by balancing bias and variance (Hastie et al., 2009). It was believed that test loss would increase monotonically after this optimal trade-off point due to overfitting, where models capture noise in the training data. As such, the optimal function class was thought to be in the undercapacity/under-parameterized regime, in which the functions cannot perfectly fit (interpolate) the training data. Regularization methods (Hastie et al., 2009) like L2, weight decay, Lasso, Early Stopping etc. are used to achieve the optimal trade-off.

## 2.2 Over-parameterization and the Second-descent Regime

The success of deep neural networks in tasks like image recognition around 2012 (Krizhevsky et al., 2012) marked a (sub-)paradigm shift inside generalization-centric machine learning. Over-parameterized neural networks, possessing more parameters than required to perfectly fit the training data (interpolation threshold), surprisingly continued to improve as they became even more over-parameterized, surpassing the performance of under-parameterized models (He et al., 2016; Neyshabur et al., 2018). This phenomenon, where increasing model complexity beyond the interpolation threshold leads to improved performance, offered a new perspective beyond the classical bias-variance trade-off. It inspired the proposal of the "double-descent curve" (Figure 3 (b)) to accommodate both the traditional U-shaped curve in the under-parameterized regime and the observed single-descent curve in the over-parameterized regime (Belkin et al., 2019).

Key questions arose: why do over-parameterized neural networks generalize well, exhibiting benign overfitting (Bartlett et al., 2020), and how can we design algorithms and architectures to improve generalization? A central question is why some global minima generalize much better than others in over-parameterized models (Zhang et al., 2021; Neyshabur et al., 2018). This led to research on implicit regularization (Neyshabur, 2017) in neural networks, aiming to provide theoretical and practical insights. Several key findings emerged:

**Large Learning Rate is Better.** Practitioners observed better generalization with near-maximal stable learning rates. Explanations include the flat minima hypothesis (SGD noise helps escape sharp minima (Keskar et al., 2016)), the edge-of-stability concept (larger learning rates decrease hessian curvature) (Cohen et al., 2020), and escaping the linearization regime (large learning rates help networks learn useful representations) (Lewkowycz et al., 2020).

**Small Batch Size is Better (Keskar et al., 2016; Smith & Le, 2017).** A critical batch size exists, below which generalization does not worsen with decreasing batch size. Above this, performance degrades, resembling a ReLU-shaped curve. This is attributed to gradient noise from small batches biasing towards flatter minima, while large batches lead to sharp minima.

These insights were developed to improve and explain generalization through implicit or explicit regularization, which is believed to reduce variance, guiding the network to find better minima (Foret et al., 2020).

Fig. 4 (a) depicts typical learning dynamics in this paradigm: training and test error curves diverge after a period of training, creating a generalization gap, while training error (classification) reaches a global minimum. Minimizing this generalization gap remains a central focus in this paradigm

We summarize two key principles in the generalization-centric paradigm:

- **Guiding Principles for Generalization**: Overfitting is a core challenge, and "regularization" serves as the primary guiding principle for understanding and improving generalization.

- **Model Comparison via Validation Set**: Due to the relatively smaller scale of problems in this paradigm, we can afford to train multiple models and rely on hold-out validation sets for model comparison, which is a reliable and effective approach.

However, in the scaling paradigm, we may lose the advantages offered by both of these principles.

## 2.3 Heavy Under-parameterization and the Skydiving Regime

Breakthroughs in large language model pretraining leads us to a scaling-centric paradigm, distinguished from the previous generalization-centric paradigm by two key features. First, the complexity of training data $\mathcal{T}$ far surpasses the capacity of the models (Raffel et al., 2020; Brown et al., 2020; Hoffmann et al., 2022; Touvron et al., 2023) and the training loss remains far from reaching a plateau. Second, both the data and the models themselves operate at a scale vastly larger than in previous paradigms, as illustrated in Figure 2 blue dot •.

Figure 4 (b) illustrates the typical learning dynamics in this paradigm: test and training error curves remain closely aligned throughout training, even when model size and compute are scaled up by factors of 500 and 250,000, respectively. The training error has not yet reached its global minimum, suggesting further scaling up either or both the model size and dataset size could lead to improved performance. This behavior corresponds to the "skydiving (blue)" regime depicted in Figure 1, preceding the U-shaped bias-variance trade-off curve (Fig. 3 (a)). In this regime, training and test errors on a holdout evaluation set are nearly identical. Consequently, the primary goal shifts from mitigating overfitting to minimizing the pretraining (approximation) error, as the generalization gap has yet to emerge.

This divergence in primary objectives suggests that conventional wisdom from generalization-centric machine learning might not readily apply to the new paradigm. Specifically, "regularization" may no longer be the driving force for performance improvement (see Section 4) nor the guiding principle for understanding scaling. Furthermore, Section 5 explores the challenges arising from the immense scale involved. In particular, the hold-out validation set method may not be applicable for model comparison in this setting.

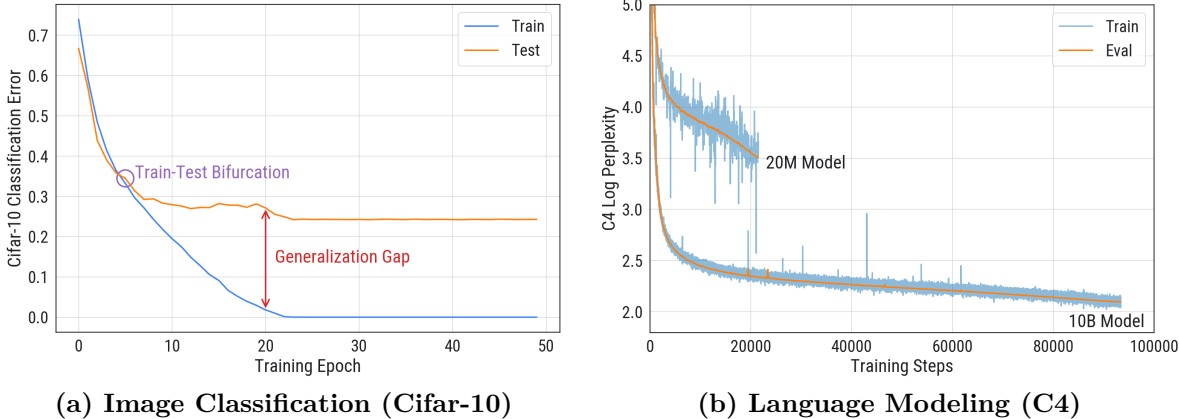

**(a) Image Classification (Cifar-10)**  **(b) Language Modeling (C4)**

Figure 4: **Learning Dynamics: Generalization (Image Classification) vs. Scaling (Language Model Pretraining). (a)** ResNet-18 on CIFAR-10. Training and test error curves initially overlap, then diverge, forming a generalization gap. Minimizing this gap is the central objective as the network easily interpolates training data. **(b)**. Decoder-only transformer on C4. Evaluation curves consistently remain within training curves throughout training, even when the model size and compute is scaled up by a factor of 500 and 250,000, respectively.

## 3 Architecture and Optimizer

We use decoder-only transformer architecture (Vaswani et al., 2017). All language models are trained on the C4 dataset (Raffel et al., 2020). We use the open-source NanoDO codebase (Liu et al., 2024) for our training process. Specific architectural details are listed below.

- **Rotary** (Su et al., 2024) Positional Embedding.

- **QK-Norm** (Gilmer J. & J., 2023; Dehghani et al., 2023), i.e., two Layer Normalization layers are applied to the queries and keys before the dot-product attention computation

- **Untying** the head from the embedding, i.e., we do not use weight tying of the first and last layer.

- **Gelu** (Hendrycks & Gimpel, 2016) activation with $F = 4D$, where $D$ and $F$ are the model dim and hidden dim of the MLP, resp. However, we use Geglu (Shazeer, 2020) in some experiments.

- The head dimension of query and key is set to $d_{\text{head}} = 64$, resulting in $H = D/d_{\text{head}}$ attention heads throughout this paper.

- The sequence length is $S = 512$.

- The vocabulary size is $V = 32101$.

- The total number of parameters in the backbone is approximately $\mathcal{N} \approx 12D^2L$, where $D$ represents the model dimension and $L$ is the number of layers in the transformer.

- Most models are trained to Chinchilla optimality (Hoffmann et al., 2022), utilizing a total of $\mathcal{D} = 20 \times (12D^2L + DV)$ tokens.

- The total compute is estimated using the "$\mathcal{F} = 6\mathcal{N}\mathcal{D}$" formula Kaplan et al. (2020), where the estimated number of floating-point operations (FLOPs) $\mathcal{F}$ is $6 \times$ Number of Parameters($\mathcal{N}$) $\times$ Number of Tokens($\mathcal{D}$).

**Optimizer.** Our default optimizer is AdamW (Kingma & Ba, 2014; Loshchilov & Hutter, 2017; DeepMind et al., 2020) with $\beta_1 = 0.9$, $\beta_2 = 0.95$, $\epsilon = $ 1e-20 and coupled weight decay $\lambda = 0.1$.

# 4 Is Regularization Needed?

As discussed earlier, regularization plays a pivotal role in the generalization-centric paradigm. It effectively mitigates overfitting and bridges the gap between training and test losses. In this section, we revisit three popular regularization techniques commonly employed in machine learning: explicit L2 regularization, and the implicit regularization effects of large learning rates and small batch sizes.

While the conclusion of this section — that explicit/implicit regularization is *not* necessary in the absence of overfitting — may be obvious to many, it is useful to re-examine our prior beliefs in the context of the scaling-centric paradigm.

## 4.1 Does L2 Regularization Improve Performance?

To assess the impact of L2 regularization across different training regimes, we compare its benefits in scaling-centric versus generalization-centric settings.

First, we showcase the usefulness of L2 in improving generalization. We trained a ResNet-50 on ImageNet using Flax's default settings, which include L2 regularization ($\lambda = 0.0001$). We then trained a second model without L2 regularization ($\lambda = 0$). As shown in Fig. 6, L2 regularization substantially boosts test accuracy by 6% (0.764 vs. 0.703). While the model without L2 achieves a higher training accuracy ($\sim 0.856$ vs. $\sim 0.807$), this clearly highlights the effectiveness of L2 regularization in reducing overfitting and improving generalization.

Next, we provide evidence that L2 regularization may not be useful for language model pretraining. To do so, we trained four 151M transformers, varying the use of L2 regularization and weight decay. The training dynamics are presented in Figure 5, leading to the following observations:

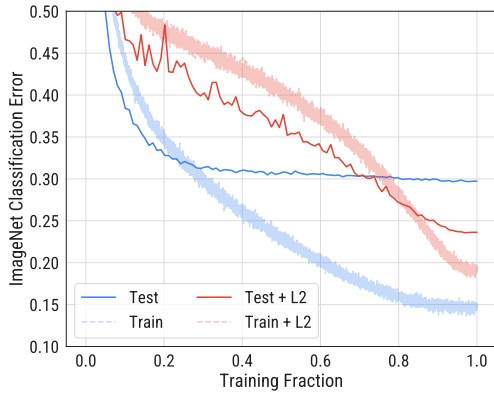

1. Operating within the "skydiving regime," where training and test losses closely track each other, implying no generalization gap, the addition of L2 regularization does not alter this behavior.

2. In our experiments, the baseline model without L2 or weight decay achieves a final evaluation loss of 2.980. Introducing L2 regularization alone (2.975) or in combination with weight decay (2.983) does not appear to improve test loss. However, employing weight decay independently leads to a notable performance gain (2.952).

Figure 6: ResNet-50 on ImageNet. L2 regularization reduces overfitting and improves generalization.

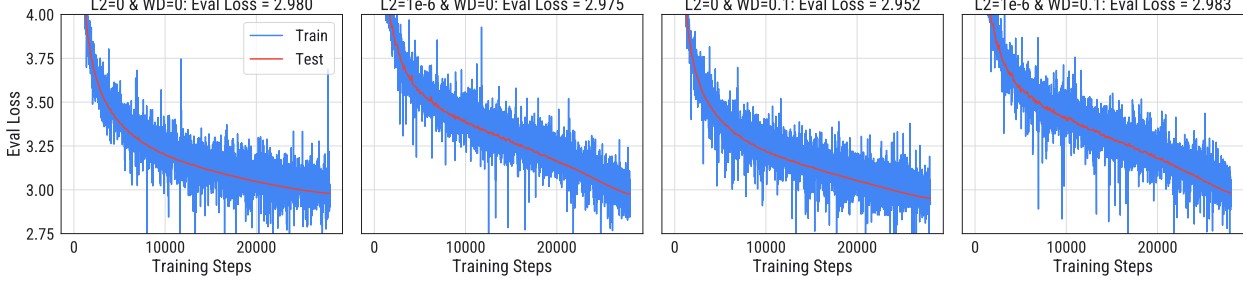

Figure 5: Training dynamics of four transformer models. From left to right: no L2 and no weight decay, small L2 and no weight decay, no L2 but with weight decay, with both L2 and weight decay.

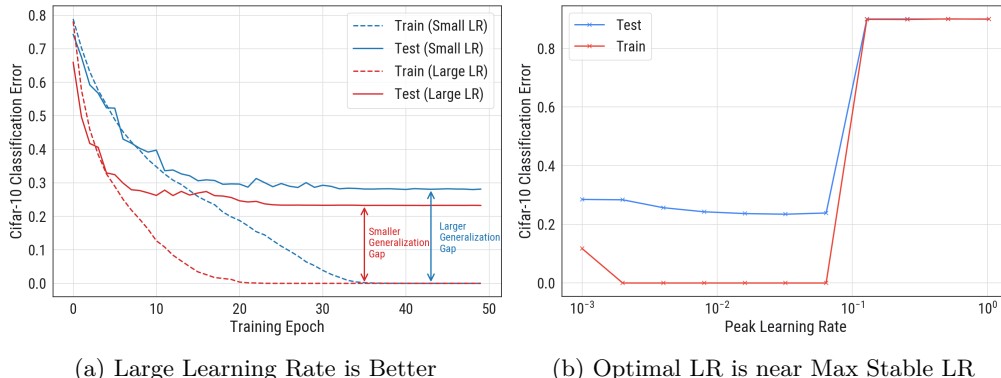

(a) Large Learning Rate is Better  (b) Optimal LR is near Max Stable LR

Figure 7: **Large Learning Rate Improves Generalization.** (a) Networks trained with large or small learning rates can achieve perfect training accuracy, but large learning rates generalize better. (b) The optimal learning rate is often near the maximum stable rate.

**Discussion.** While L2 regularization is widely acknowledged to reduce overfitting and thus enhance generalization performance, our preliminary experiments indicate that it may not offer similar benefits for language model pretraining. This aligns with current practices, as flagship language models such as GPT-3 (Brown et al., 2020), PALM (Chowdhery et al., 2023), Chinchilla (Hoffmann et al., 2022), Llama-2 (Touvron et al., 2023) and DeepSeek-V2 (DeepSeek-AI et al., 2024) do not employ L2 regularization. While weight decay is widely used in training language models, our observations are consistent with Andriushchenko et al. (2023) in that it does not play a conventional regularizer role. Guiding principles are needed to deepen our understanding of weight decay in this context.

### 4.2 Does Maximal Stable Learning Rate Perform Better?

Conventional wisdom in neural network training often favors using a larger learning rate, possibly near the maximum stable value, as this is believed to improve generalization performance (Li et al., 2019; Lee et al., 2020; Lewkowycz et al., 2020). This practice is attributed to the implicit regularization of stochastic gradient descent (SGD). While various learning rates can achieve perfect training accuracy for small datasets, e.g. CIFAR-10, the gradient noise from larger learning rates is thought to guide SGD towards better minima. This phenomenon has been explored in several influential studies, including those on flat minima (Hochreiter & Schmidhuber, 1997; Keskar et al., 2016; Dinh et al., 2017; Foret et al., 2020), the edge of stability (Cohen et al., 2020; Agarwala et al., 2022; Damian et al., 2022; Gilmer J. & J., 2023), and escaping the NTK regime (Jacot et al., 2018; Chizat et al., 2019; Lee et al., 2019; Lewkowycz et al., 2020; Yang & Hu, 2021; Woodworth et al., 2020; Karkada, 2024). Although no rigorous theory fully explains why the maximal stable learning rate improves performance, it's considered by many a good practice in training neural networks. We reproduce this insight by training a ResNet-18 on CIFAR-10. Figure 7a shows the regularization benefits of using a larger learning rate: while both large and small learning rates lead to zero classification error, the larger learning rate results in a smaller test error. Fig. 7b shows that the optimal learning rate is near the maximal stable learning rate.

Does this conventional wisdom apply to training language models? Since the primary principle behind this practice is improving generalization (via implicit regularization) - different from the primary goal in the scaling-centric paradigm - we might expect it to not hold true. In the following, we provide empirical evidence supporting this hypothesis.

**Experiment Setup.** To investigate the optimal learning rate for training language models, we trained our base model to Chinchilla-Optimal while varying the learning rates. We further test the robustness of our findings by introducing several interventions:

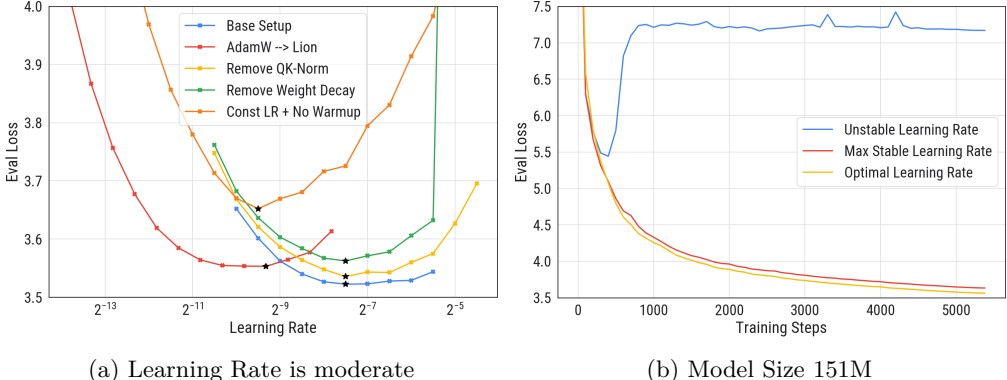

(a) Learning Rate is moderate       (b) Model Size 151M

Figure 8: **Optimal Learning Rates Are Significantly Lower Than Maximal Stable Learning Rates.** (a) Loss vs. learning rate curves reveal U-shaped relationships, with optimal learning rates far below stability limits. (b) Despite smooth training curves, maximal stable rates consistently underperform.

1. Switching from AdamW to Lion (Chen et al., 2024) (Fig. 8a),

2. Removing QK-Norm (Fig. 8a),

3. Eliminating weight decay (Fig. 8a),

4. Removing warmup and learning rate schedule (Fig. 8a),

5. Adjusting batch size and changing model size (Fig. 10a and Fig. 10b)

**Experimental Results.** Across all these setups, the relationship between loss and learning rate consistently exhibited a U-shape curve. The optimal learning rate was far from the maximal stable learning rate often favored in traditional neural network training.

**Discussion.** Contrary to conventional wisdom, our findings reveal that the optimal learning rate for large language models is significantly lower than the maximal stable value[3] previously assumed. This suggests that traditional regularization-based theories may not fully explain the dynamics of optimal learning rates in the context of training language models. Further research is needed to elucidate the complex relationship between optimal learning rate, model scale, and other training factors (see Paquette et al. (2024)).

### 4.3 Does Small Batch Size Perform Better?

In generalization-centric ML, a common observation is that, given the same computational budget (measured by the total number of training epochs), algorithms employing smaller batch sizes tend to generalize better than those with larger batches beyond a critical threshold (Keskar et al., 2016; Smith & Le, 2017; Shallue et al., 2019; McCandlish et al., 2018). This phenomenon is often attributed to the increased gradient noise associated with smaller batches, which acts as an implicit regularizer, mitigating overfitting. It's hypothesized that the noise helps guide the optimization process towards minima that generalize better.

We replicate this conventional wisdom in Fig. 9 using the ImageNet example in (Heek et al., 2023). With a fixed number of training epochs (100), the trend demonstrates that smaller batch sizes lead to better test performance, albeit with worse training performance, effectively reducing overfitting.

However, does this conventional wisdom about small batch sizes translate to the realm of LLM training? Given that the primary benefit of small batches is their regularization effect, and regularization may not be the primary concern for LLMs, we have reason to question this assumption.

---

[3]This observation has been recognized (implicitly or explicitly) in existing work, e.g. Wortsman et al. (2023); Zhao et al. (2024) among many others.

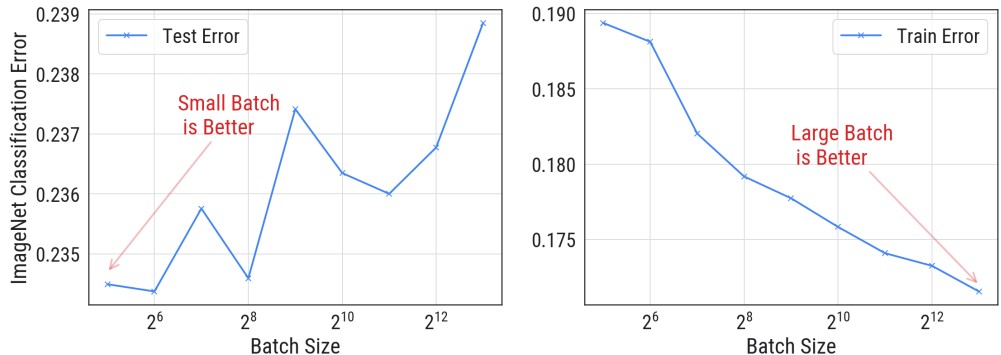

Figure 9: **Small Batch Size performs better for ImageNet.** Small batch size achieves better *Test Error* (left) but worse *Training Error* (right), vice versa for large batch size.

**Experiment Setup.** We trained two models (19M and 151M parameters) to Chinchilla-Optimal performance, varying batch sizes in $\{16, 32, 64, 128, 256, 512, 1024\}$. For each batch size, an 11-point grid search was performed to identify the optimal learning rate. To assess variability, the 19M model was trained with five different random seeds.

**Experimental Results.** Fig. 10a and Fig. 10b illustrate the evaluation loss as a function of learning rate for various batch sizes. Aggregating the best evaluation loss for each batch size, we observe a clear U-shaped curve in Fig. 10c. This demonstrates that both excessively small and large batch sizes can negatively impact model performance.

**Discussion.** The conventional wisdom that smaller batch sizes lead to better performance may not always apply to language model pretraining. While gradient noise from small batch sizes can potentially benefit generalization, it may also impede optimization. The observed U-shaped relationship between loss and batch size raises interesting questions. Why does this U-shape occur, and what trade-offs determine the optimal batch size? These questions remain open for further investigation and could provide valuable insights into improving the efficiency of large-scale language model training.

## 4.4 Discussion

Through three examples, we provided evidence that regularization, either implicit or explicit, may not be necessary for language model pretraining. It may no longer be the main driving principle for understanding pretraining or making informed decisions during training. This raises a crucial question: what are the emerging guiding principles in the scaling-centric paradigm? See Section 6 for a more in-depth discussion.

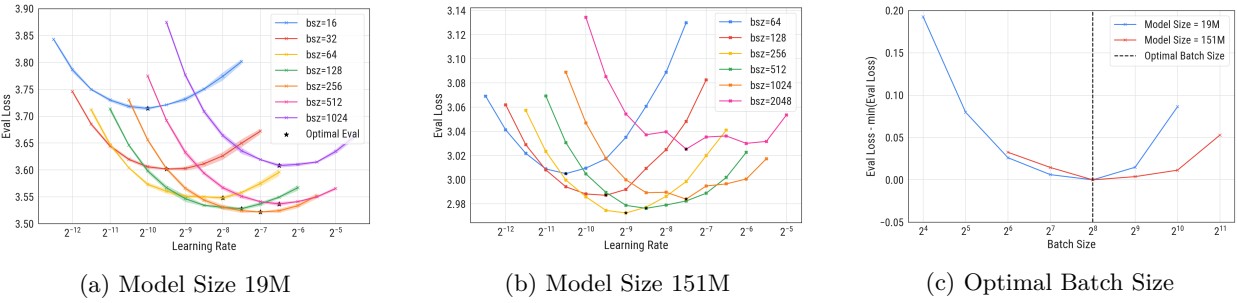

(a) Model Size 19M  (b) Model Size 151M  (c) Optimal Batch Size

Figure 10: **Optimal Batch Size.** Learning rate search for various batch sizes. (a) 19M model, (b) 151M model. (c) Performance as a function of batch size is U-shaped and is sensitive to the choice of batch size. Both large and small batch sizes lead to sub-optimal performance.

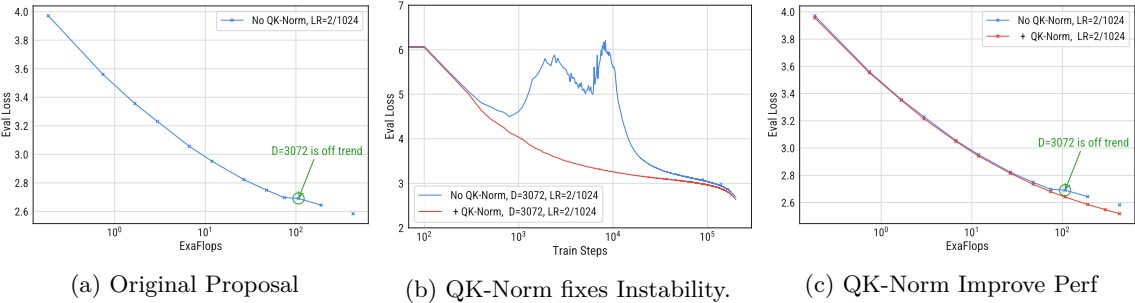

(a) Original Proposal     (b) QK-Norm fixes Instability.     (c) QK-Norm Improve Perf

Figure 11: **Training Instability.** (a) The model with $D = 3072$ deviates from the expected scaling trend. (b) Closer examination of the training dynamics reveals that the $D = 3072$ model exhibits instability. This instability is mitigated by the application of QK-Norm. (c) With QK-Norm, the scaling law exhibits a normal and smooth behavior.

## 5 Scaling Law Crossover, a Curse from Scale?

In the generalization-centric paradigm, the scales we operate on are significantly smaller than those in the LLM setting. This allows us to test new ideas on smaller datasets like CIFAR10 and, if successful, scale them up to ImageNet. Since hyperparameter tuning is feasible even at ImageNet scales, we can readily test our ideas without excessive concern about hyperparameters. Consequently, when proposing new ideas, we often pay little attention to how hyperparameters should evolve with scale, assuming users can perform the tuning themselves. For instance, when introducing alternative architectures (e.g., skip connections, batch normalization), data augmentation techniques, or new optimizers like AdamW, we don't necessarily need to provide guidance on how the learning rate and other hyperparameters should scale with model and dataset sizes.

However, in the "skydiving" regime, the sheer scale of data and models presents a significant challenge. Traditional hyperparameter tuning becomes impractical due to the immense computational costs involved. This makes it incredibly difficult and expensive to verify new ideas or compare different approaches at scale. Consider the seemingly simple question of whether a global gradient clipping norm of 1 or 2 is more effective in a 100B parameter model. Answering this question through direct experimentation would require substantial resources and time, highlighting the unique challenges posed by this regime.

In what follows, we present a new phenomenon, termed *scaling-law crossover*, where the effectiveness of different techniques reverses at a certain scale. One idea outperforms another below a critical scale, while the opposite holds above it.

We present three cases of scaling-law crossover with increasing complexity, offering explanations for the first two but leaving the third as an open question that underscores the complexities of scaling. Unlike the traditional "test on CIFAR, scale to ImageNet" workflow, the reality of scaling laws may necessitate a continuous climb up the scaling ladder: testing at progressively larger scales until a crossover is observed, or we gain enough confidence to bet that the idea is effective at the desired scale. Therefore, evaluating an idea's potential may take days or even weeks, and require 100+ GPUs and a group of researchers. This process becomes increasingly harder and more costly as we climb up the scaling ladder.

### 5.1 Warmup: Training Instability

Training instability in large language models (LLMs) is a widely recognized challenge within the research community (Liu et al., 2020; Chowdhery et al., 2023; Dehghani et al., 2023; Zhang et al., 2023; Molybog et al., 2023; Cohen et al., 2022; Wortsman et al., 2023). In this section, we reproduce this phenomenon and explore potential solutions. While our initial fix appears to resolve the issue, it hides a deeper problem in scaling strategies.

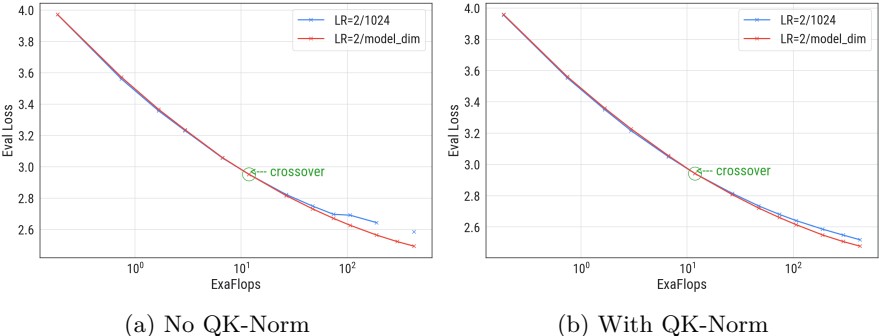

(a) No QK-Norm

(b) With QK-Norm

Figure 12: **Constant Learning Rate vs LR $= 2/D$.** A constant learning rate LR $= 2/1024$ and a learning rate of LR $= 2/D$ yield nearly identical performance at small scales ($D \leq 1024$). However, their performance diverges at larger scales ($D > 1024$), with LR $= 2/D$ demonstrating superior performance both (b) with and (a) without QK-Norm.

**Experiment Setup.** We train a series of models with varying dimensions $(D, F) = (128k, 512k)$ for $k \in [1, 2, 3, 4, 6, 8, 12, 16, 20, 24, 32, 40, 48]$. All models share the same batch size (256), number of layers (6), sequence length (512), and training steps (200,000, not Chinchilla-Optimal). We use a constant learning rate LR $= 2/1024$ for all model sizes.

In Figure 11a, we plot the evaluation loss against computational cost (flops) for each model. While performance generally improves with scale, a discontinuity emerges around $10^2$ exaflops ($D = 3072$). Examining the learning dynamics for $D = 3072$ (Figure 11b) reveals training instability in larger models: the evaluation loss spikes during training (between 1,000-10,000 steps) but self-corrects later. Clearly, our scaling approach requires adjustment.

Fortunately, several techniques exist to address training instability: Z-loss (Chowdhery et al., 2023), extended warmup periods (Wortsman et al., 2023), learning rate reduction, increased weight decay, and QK-Norm (to prevent attention logit explosion (Dehghani et al., 2023)).

We try QK-Norm, and it proves effective. Not only does it resolve the instability for $D \geq 3072$ (Figure 11b), but it also enhances performance for larger model sizes (Figure 11c). However, a crucial question remains: have we truly addressed the underlying issue, or have we merely patched a symptom?

## 5.2 Sub-Optimal Learning Rate Scaling Rule

Years of research investment in understanding the training dynamics of neural networks has taught us a valuable lesson: learning rates should be adjusted based on training specifics, particularly model scale Goh (2017); Lee et al. (2019); Sohl-Dickstein et al. (2020); Xiao et al. (2019); Yang et al. (2022); Bi et al. (2024). The challenge lies in determining how to adjust them effectively.

We compare two proposals: using a constant learning rate (LR $= 2/1024$ as in Section 5.1) versus scaling the learning rate with model width (LR $= 2/D$) Everett et al. (2024).

Proposal Blue. Constant learning rate LR $= 2/1024$ for all model sizes.

Proposal Red. Learning rate scales with model dimension, LR $= 2/D$.

We evaluate these proposals with and without QK-Norm. Results are presented in Figure 11.

**Observations.**

1. Using $2/D$ learning rates mitigates training instability, even without QK-Norm.

2. While the two proposals yield nearly identical performance at small scales ($D \leq 1024$), with Proposal Blue potentially slightly better, a crossover occurs at $D \geq 1024$. Beyond this point,

Proposal Red consistently outperforms Proposal blue, and this performance gap widens with increasing scale, regardless of the presence or absence of QK-norm (Fig. 12a and 12b.)

Retrospectively, the compute inefficiency of Proposal Blue can be attributed to sub-optimal learning rate or sub-optimal hyperparameter choices. However, these inefficiencies may not be apparent until we identify the sub-optimal component in our proposal and discover a new one to fix it (in this case, changing learning rate scaling rule from constant to $2/D$). We might have mistakenly believed that adding QK-norm to Blue fully addresses the issue, unaware of deeper problems within our proposals. This raises three questions:

- Is the $2/D$ learning rate scaling rule a true cure or merely a band-aid solution? Could there be an even better one?

- Are there inherent limitations or hidden bottlenecks in our model architectures that degrade performance at large scales but are not observable at small scales?

- How can we effectively distinguish between proposals whose performance is nearly indistinguishable at small scales, yet exhibits significant differences at larger scales that are too expensive to run?

### 5.3 Sub-optimal Weight Decay Scaling Rule

**Experiment Setup.** The architectures remain consistent with previous sections, varying only the scaling factor $k$ in $(D, F) = (128k, 512k)$. Crucially, models are trained to Chinchilla-Optimal, meaning the training horizon scales with model size (20 tokens per parameter).

Proposal Blue: We employ muP (Maximal Update Parametrization (Yang et al., 2022)) and a constant, dimension-independent weight decay. Optimal hyperparameters were determined through hyperparameter search with independent weight decay ($\lambda = 0.000566$) which remains the same for all $D$, and learning rate ($\eta = 0.0055$) for the base model $D = 512$.

Proposal Red: We scale the learning rate inversely with the model size (LR $= 2/D$), and additionally, the weight decay is scaled similarly, with independent weight decay $\lambda = 0.1 \times 2/D$.

Note that Proposal Blue is consistent with the recommendation from the muP paper "weight decay should scale independently with width" (Yang et al., 2022; 2023).

**Observation.** While muP is often considered a best practice for scaling up models, its performance gains observed at smaller scales may not translate effectively to larger ones. A crossover point emerges, beyond which Proposal Blue (based on muP) loses its advantage; see Fig. 13a. This is likely because muP assumes that the number of training tokens (and thus, training steps) is constant with respect to model dimension ($D$). However, this assumption doesn't hold in practice, especially with Chinchilla-Optimal scaling, which suggests a training token count of $20\mathcal{N}$ (where $\mathcal{N}$ is the number of parameters, proportional to $D^2$ for fixed layers). Therefore, muP's underlying assumption conflicts with the Chinchilla-Optimal scaling strategy (Everett et al., 2024). As shown in Fig. 13b, constant weight decay leads to significant suppression of parameter norms throughout training. Scaling the weight decay with the learning rate (LR $= 2/D$) results in better parameter norm dynamics.

### 5.4 Is Gradient Normalization a Good Idea?

**Experiment setup.** In this experiment, we co-scale model dimension $D$ and number of layers $L$ with a fixed aspect ratio $D/L = 512/6$ and train all models to Chinchilla-Optimal. Our learning rate search procedure suggests the formula $\eta = 2/D \times (6/N)^{0.675} \times 2^{0.25}$. We call this Proposal Red.

Proposal Blue: We propose the following modifications to the base model:

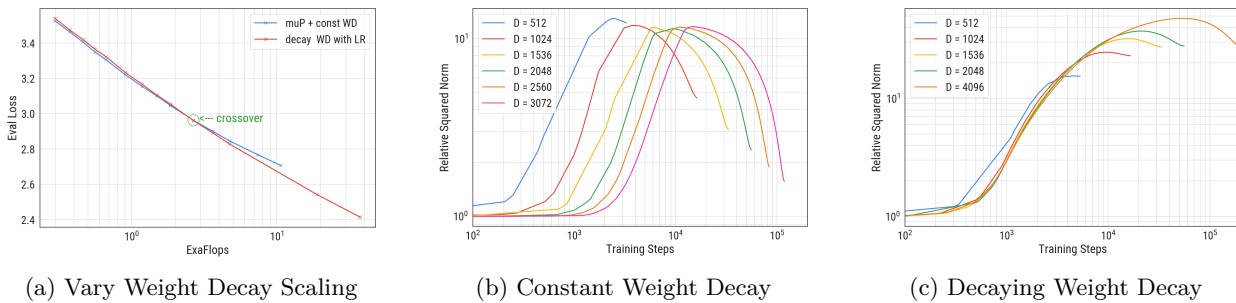

(a) Vary Weight Decay Scaling   (b) Constant Weight Decay   (c) Decaying Weight Decay

Figure 13: **Crossover Phenomenon due to Sub-optimal Weight Decay.** (a) A constant weight decay strategy initially outperforms a decaying weight decay approach at smaller scales. However, this advantage diminishes and reverses at larger scales, demonstrating a crossover phenomenon. (b) With constant weight decay, parameter norms are significantly suppressed throughout training. (c) Decaying the weight decay alongside the learning rate results in less suppressed parameter norms.

1. **Activation Function and MLP Dimension:** Replace the GeLU activation function with its gated variant, GeGLU (Shazeer, 2020), and increase the hidden dimension of the MLP to $F = 6D$. Gated activations like GeGLU are commonly believed to improve model performance.

2. **Weight Decay:** Increase the weight decay by a factor of 4.

3. **Gradient Normalization:** Normalize the raw gradients by their root mean square (RMS) before passing them to the AdamW optimizer.

The ablation study (Fig. 14a) shows promising results for these modifications, particularly in terms of improved performance and reduced learning rate sensitivity (Wortsman et al., 2023). It seems we may have found an innovation. Let's now evaluate Proposal Blue at scales.

**Observation.** Evidently, we encounter another scaling law crossover, as illustrated in Figure 14b. While Proposal Blue initially exhibits promising results at smaller scales, this advantage diminishes, leading to a crossover point around $2 - 3 \times 10^2$ exaflops — a significantly larger scale than observed in previous experiments. Unlike those instances, the cause of this crossover remains elusive. Does gradient normalization become inherently detrimental at larger scales? Should the learning rate scaling formula be adjusted when

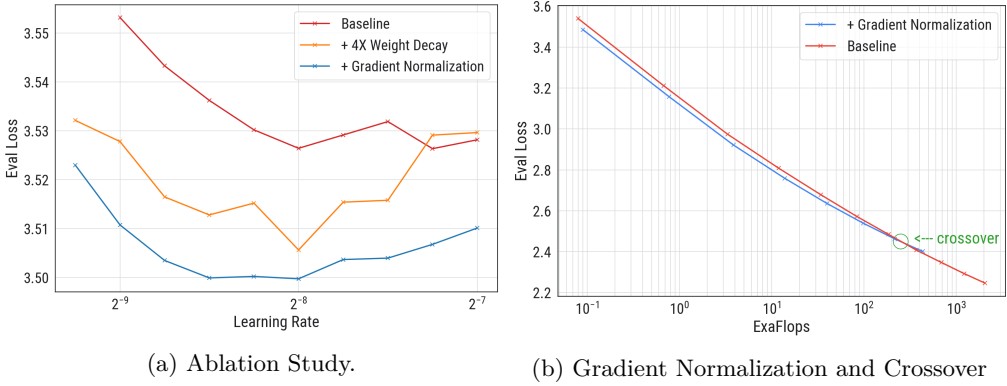

(a) Ablation Study.     (b) Gradient Normalization and Crossover

Figure 14: **Gradient Normalization Leads to Scaling Law Crossover?** (a) An ablation study showing Proposal Blue performs better at small scales. In particular, gradient normalization not only improves performance but also reduces learning rate sensitivity in the scale we are testing. (b) Performance gain from Proposal Blue does not transfer to large scale. A crossover occurs around $2 - 3 \times 10^2$ exaflops.

incorporating gradient normalization? Even if we successfully address the limitations of the gradient normalization proposal and verify its effectiveness up to $10^3$ exaflops, can we confidently assert that the fix will remain effective at even larger scales, such as $10^4$ exaflops?

### 5.5 Discussion

We discuss several direct consequence of scaling law crossover and leave a more in-depth discussion for the next section.

1. **Hyperparameter Tuning at Scale.** As performance is highly sensitive to the choice of hyperparameter scaling rules, (False Negative) good ideas may be killed owing to insufficient hyperparameter tuning and (False Positive) sub-optimal ideas may be promoted due to weak baseline.

2. **Credit Assignment.** The crossover scaling phenomenon underscores that demonstrating impressive performance at small scales is insufficient. While proposing new ideas and testing them at small scales remains crucial, rigorously verifying ideas at large scales demands substantial effort, resources, and, crucially, faith in their potential. Thus, we argue that scaling up existing ideas and rigorously demonstrating their effectiveness at scale is as important as, or even more important than, proposing new ideas and testing them on small scales. Both types of contributions are essential and should be recognized and valued.

3. **Avoid Biased Search Spaces.** The scaling law crossover phenomenon indicates that it is crucial to avoid overemphasizing ideas that work well at small scales. This narrow focus might lead us to miss groundbreaking approaches, similar to the "vision transformer" (Dosovitskiy et al., 2020), that excel at large scales but might not shine on smaller ones.

## 6 Discussion and Conclusion

Let us revisit the two central themes: model comparison at scale and guiding principles for scaling.

### 6.1 Model Comparison at Scale

The ability to effectively and reliably compare models is fundamental to advancing machine learning. In the generalization-centric paradigm, the validation set approach remains a *simple*, *reliable*, *cost-effective* and *theoretically grounded* method for such comparisons. However, this approach does not apply to the scaling-centric paradigm. The immense scale often prevents training multiple models for comparison. Furthermore, the phenomenon of scaling law crossover — where the relative performance of methods can change as models scale — poses a fundamental challenge: we cannot simply compare two models at small scales and assume that the observed ranking will hold at larger scales. It raises a fundamental question:

**Model Comparison at Scale**: How to compare models at a scale where training is feasible only once?

We discuss two possible methods below.

### 6.1.1 Scaling Law Extrapolation for Model Comparision

The first approach relies on scaling law extrapolation (Kaplan et al., 2020): extrapolating observations from smaller scales to predict performance at larger scales. Specifically, we assume the following functional form relating loss $\mathscr{L}$ to computational cost (flops, denoted by $f$):

$$\mathscr{L}(f) = af^b + c \tag{4}$$

for a given class of models (Kaplan et al., 2020), where the parameters $(a, b, c)$ depend on the model's specific characteristics. We then generate a sequence of measurements $\{(f_i, l_i)\}_{1 \leq i \leq k}$ by training a series of models up to a certain scale (e.g., $k = 5$ and up to 40 exaflops, as shown in Fig. 15), where $(f_i, l_i)$ represents the flops and loss of the $i$-th model. Next, we find the optimal values of $(a, b, c)$ that best fit these

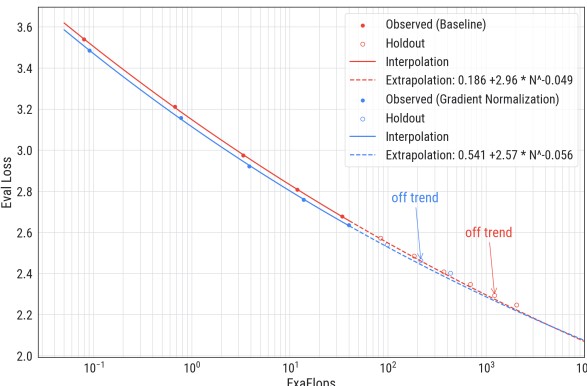

Figure 15: **Scaling Law Extrapolation**: Naively fitting a power law to observed data points can lead to inaccurate extrapolations. For example, our "Blue" and "Red" proposals showed significant deviations after 10x and 30x compute extrapolation, respectively.

measurements. Finally, we use Equation 4 to make predictions about future performance, such as the expected loss at $f = 5e3$ exaflops.

We apply this approach naively to Proposal Blue and Proposal Red from Section 5.4, using $k = 5$ and $f \leq 40$ exaflops. The results are presented in Figure 15.

Although our scaling law extrapolations capture the overall scaling trend, the precision of the predictions is insufficient for reliable model comparison. Proposal Blue deviates from the predicted trend around $3 \times 10^2$ exaflops, failing to achieve a 10x extrapolation, while Proposal Red deviates around $10^3$ exaflops, failing a 30x extrapolation. Furthermore, the scaling law extrapolations incorrectly predict that crossover occurs in the interval $[4 \times 10^3, 5 \times 10^3]$, while the actual crossover occurs in $[2 \times 10^2, 3 \times 10^2]$, as shown in Figure 14b. This demonstrates that naively extrapolating scaling laws for model comparison can be unreliable and lacks a solid theoretical foundation. In contrast, the GPT-4 technical report (OpenAI et al., 2023) showcases the potential for accurate 1,000x and even 10,000x extrapolations, although the specific techniques and conditions enabling such accurate extrapolations remain unclear.

Overall, to reliably apply scaling law extrapolation for model comparison at scale, we believe extensive research is necessary to fully comprehend both the macro and micro dynamics at play. This may include a deeper understanding of the intricate relationship between optimization, architectures, data, and scales (Kaplan et al., 2020; Bahri et al., 2024; Hoffmann et al., 2022; DeepSeek-AI et al., 2024; Paquette et al., 2024; Bordelon et al., 2024; Lin et al., 2024), as well as the subtleties within our machine learning systems. These subtleties encompass factors such as learning rate schedules (Hoffmann et al., 2022), parameter counting choices, the number of warmup steps (Porian et al., 2024), curve fitting approaches (Besiroglu et al., 2024), and even the epsilon value in AdamW (Wortsman et al., 2023; Everett et al., 2024).

### 6.1.2 Are Hyperparameter Transfer sufficient for Model Comparison?

$\mu$-Transfer (Yang et al., 2022) is an important technique for tuning hyperparameters in large models. Its core idea involves parameterizing the network (using maximal update parameterization) and scaling the learning rate appropriately, enabling the zero-shot transfer of optimal hyperparameters (e.g., learning rates, scale of initialization) from small models to much larger ones. While these methods have demonstrated promising results (Yang et al., 2022; Lingle, 2024; Blake et al., 2024; Everett et al., 2024), certain limitations hinder their direct application to model comparison.

Firstly, hyperparameter comparison represents only a small fraction of model comparison, which encompasses optimization choices (e.g., AdamW vs. Lion, schedule-free optimizer, Shampoo (Chen et al., 2024; Defazio et al., 2024; Gupta et al., 2018)), architectural variations (e.g., multi-head attention vs. grouped-query attention (Ainslie et al., 2023; Shazeer, 2019), transformers vs. non-transformers (Gu & Dao, 2023;

Botev et al., 2024; Beck et al., 2024)), and data considerations (e.g., different data mixtures (Penedo et al., 2024), varying token-to-parameter ratios).

Secondly, the current $\mu$P framework primarily focuses on scaling width[4] while maintaining other factors like batch size, the number of layers, and training steps as static hyperparameters. This deviates from practical scenarios where batch size, depth, width, and training steps often co-evolve with scale. Empirical evidence suggests that hyperparameters may not transfer seamlessly when scaling more than one dimensions concurrently (Everett et al., 2024).

Thirdly, we may seek quantitative model comparisons rather than just qualitative assessments, where hyperparameter transfer may not be directly applicable. For instance, to reduce inference costs, we might overtrain a small model significantly beyond the Chinchilla-Optimal point and employ grouped-query attention instead of multi-head attention. In this scenario, we would like to quantify and then optimize the computational trade-offs between the overtrained grouped-query model and a Chinchilla-Optimally trained multi-head model.

In conclusion, hyperparameter transfer technique alone at its current form is not sufficient to resolve the challenge of model comparison at scale.

## 6.2 Guiding Principles for Scaling

In the pursuit of models that generalize well, regularization plays a central role. It serves as a guiding principle for understanding machine learning algorithms, making informed decisions during training, and inspiring novel ideas. Through the lens of regularization, we can understand a variety of phenomena and discover new techniques in machine learning. We list several examples below:

- **Hyperparameter Choices**: We can grasp the impact of hyperparameters like learning rate and batch size on generalization. For instance, we understand why larger learning rates can be beneficial and why excessively large batch sizes can hinder generalization. This understanding has inspired novel techniques like Sharpness-Aware Minimization (SAM) Foret et al. (2020) which explicitly promote generalization by seeking flatter minima in the loss landscape.

- **Weight Decay**: We can explain the regularization effects of weight decay and understand its various mechanisms for improving generalization Zhang et al. (2018).

- **Double Descent and Over-parameterization**: We can understand the regularization effect of over-parameterization and how it helps mitigate overfitting in certain regimes (Hastie et al., 2022; Mei & Montanari, 2022; Adlam & Pennington, 2020).

- **Weight Sharing and Pooling**: Weight sharing in convolutional neural networks (CNNs) induces equivalences in the function space, and global average pooling further enforces invariance. These properties provably enhance generalization by reducing (regularizing) the complexity of the function class (Mei et al., 2021).

- **Locality and Hierarchy**: Locality (e.g., local receptive fields) regularizes the network by prioritizing the learning of local interactions (simpler) before long-range ones Misiakiewicz & Mei (2022); Favero et al. (2021). Hierarchy, coupled with locality, allows models to learn complex functions in a balanced manner, capturing both local (higher-order interactions) and global (lower-order interactions) information Xiao (2022). These properties help explain why Convolutional Neural Networks (CNNs) often generalize better than Multi-Layer Perceptrons (MLPs), and why deeper CNNs often outperform shallower ones.

However, when it comes to scaling, the guiding principles become less clear. One foundamental challenge is the existence of the scaling law crossover phenomenon. This phenomenon makes it unwise to optimize

---

[4]While there has been some progress in depth scaling (Yang et al., 2023; Bordelon et al., 2023), the signal for hyperparameter transfer remains convoluted.

performance for any static scale, as such an approach may only be effective up to a certain point and fail to generalize to larger scales, i.e., it risks overfitting to a finite scale. Unfortunately, scaling law crossover is likely unavoidable in practice. Any observed scaling law represents a specific, and likely suboptimal, trajectory through a vast space of possible scaling strategies.

To illustrate this, consider training a sequence of transformers with increasing flop budgets $f \in \mathbb{N}$. We can approximate the flops as $f \approx 6\mathcal{N}\mathcal{D}$, where $\mathcal{N}$ is the number of parameters and $\mathcal{D}$ is the number of data tokens. The loss, $\mathcal{L}$, depends on numerous factors including width ($D$), layers ($L$), $\mathcal{D}$, batch size ($B$), learning rate ($\eta$), weight decay ($\lambda$), and others. For simplicity, let's focus on these six, recognizing that $\mathcal{N} \approx 12D^2L$ for transformers with $F = 4D$.

With the constraint $f = 6 \cdot 12D^2L \cdot \mathcal{D}$, we have five free variables. A scaling rule, $\phi$, dictates how these variables scale with $f$:

$$\mathcal{SR} = \{\phi : f \in \mathbb{N} \mapsto (D, L, \mathcal{D}, B, \eta, \lambda) \in \mathbb{N}^4 \times \mathbb{R}_+^2 \quad \text{with} \quad f = 72D^2L\mathcal{D}\} \tag{5}$$

Each $\phi$ produces a scaling law curve within a 5-dimensional surface. Unless we identify the optimal scaling rule(s), $\phi^*$, that minimize $\mathcal{L}$ for all $f$:

$$\mathcal{L}(\phi^*(f)) = \inf_{\phi \in \mathcal{SR}} \mathcal{L}(\phi(f)),$$

any given scaling law can likely be crossed by another. Since practical scaling rules are often heuristic and unlikely optimal, crossover is to be expected. For example:

- We did not realize that Kaplan's scaling rule $\mathcal{D} \propto f^{0.27}$ (Kaplan et al., 2020) was sub-optimal until the discovery of a better scaling rule $\mathcal{D} \propto f^{0.5}$ (Hoffmann et al., 2022). But how do we know that the Chinchilla scaling rule is optimal? Indeed, the exponent $\alpha = 0.5$ is not universal and depends on the dataset in a complicated manner (Paquette et al., 2024). In practice, a Chinchilla-type empirical analysis is often needed to determine the optimal scaling relationship between $\mathcal{D}$ and $\mathcal{N}$ for new datasets (Bi et al., 2024; Dubey et al., 2024).

- Section 5.2 shows that a constant learning rate scaling rule is sub-optimal, as scaling the learning rate with $2/D$ leads to better scaling (Yang et al., 2022; Everett et al., 2024). But how do we know there isn't an even better scaling rule?

- Section 5.3 demonstrates that a constant, independent weight decay is sub-optimal, since decaying weight decay with the learning rate leads to better scaling. Again, how do we know there isn't a better approach?

Therefore, there is little hope of identifying truly optimal scaling rules $\phi^*$ in practice. It is more likely that we will gradually identify better practices and better methodologies for scaling. To do so effectively and reliably, we need guiding principles to navigate the complex, high-dimensional scaling space $\mathcal{SR}$[5] and facilitate meaningful comparisons between different scaling strategies.

This leads to a central question in scaling:

*"What are the guiding principles for scaling that enable model comparison at scale?"*

## 7 Limitation.

Machine learning is a rapidly evolving field, and our current understanding of scaling phenomena remains limited. What holds true today may not be valid in a few months. Notably, we mainly focus on reducing

---

[5]In practice, the dimensionality of the scaling space is much greater than what is assumed here.

the loss in pretraining and our analysis assumes a "skydiving" regime, where data complexity significantly exceeds model complexity. This assumption may break in at least two settings. First, in post-training (e.g., instructional finetuning), the number of tokens are smaller than the number of parameters and reducing overfitting is important. We didn't dive into post-training in this paper. Second, as computational resources grow exponentially while high-quality data may plateau, it is likely that we will re-enter a U-shaped regime or even the second-descent regime (Fig. 1). In this case, traditional wisdom may regain relevance, and "new wisdom" may become outdated.

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
