# OpenReview forum: "Rethinking Conventional Wisdom in Machine Learning: From Generalization to Scaling"
_TMLR — Withdrawn by Authors_

### Review · Reviewer_dF7y · 2025-07-23

**Summary Of Contributions:**

The authors highlight some challenges related to the data-intensive, underparametrized regimes in which modern LLMs operate. They first highlight how conventional wisdom from traditional deep learning cease to hold in this regime. In particular, they illustrate on three examples how the effect of L2 regularization, learning rate, and batch size can drastically differ between an image-classification task and a larger scale transformer pretraining tasl. Secondly, they illustrate how two models differing by hyperparameter scaling /training strategies can compare differently at small and large compute, evidencing the challenge that strategies working well at smaller scale may not seamlessly transfer at larger scale.

**Audience:**

Yes

**Audience Explanation:**

I believe the paper offers in its first part an interesting overview of a selection of phenomenological differences between the regime characterizing modern language tasks and more classical e.g. image classification tasks. Section 5 illustrates a highly relevant challenge that behaviors at large compute may differ from that at small compute, and I believe will be, to the best of my evaluation, interesting for the TMLR readership. Therefore, I would tend towards acceptance. On the other hand, I have limited knowledge of the related literature, so cannot fully judge of the novelty of the findings.

**Claims And Evidence:**

No

**Claims Explanation:**

Overall, the plots illustrate convincingly the points and intuition discussed, and model specifications are discussed in detail in section 3. I however have a number of reserves regarding the following points:
- Most of the points are illustrated by a single setup (for instance, the discussion of the effect of L2 regularization is based on a comparison between a ResNet experiment on ImageNet (Fig. 6), and the considered transformer task (Fig. 5) solely.) These do not make for a thoroughly convincing case that the discussed difference hold in general beyond the specific example given. Plots for more models and tasks would bolster the intuition given.
- Some plots, such as Fig. 9, seem to display a single trial and are very noisy as a result. Averaging the datapoints over initialization and training, and providing the error bars, would help with the clarity of the plot, and make them more convincing.
- Some caption would gain to be further detailed. For instance, the current writing leaves it unclear whether Fig. 10 (c) is for the optimal learning rate, and if not, which learning rate was used.

I have the following concerns/questions regarding some of the conclusions, which I hope the authors may help clarify:
- In Fig. 15, the authors argue that the extrapolation does not quantitatively fit the experimental points, and wrongly predicts the range in which the crossover occurs. On the other hand, it does predict the occurrence of the crossover. Is this the case also for the other examples provided beyond that of Fig. 14, e.g. for Fig. 13?
- In the first part of the paper, the authors compare the "generalization-centric" paradigm with the "sky-diving regime". Do their findings and observed phenomenology for the latter, observed in their transformer experiments, also appear for more standard tasks (e.g. the ResNet learning ImageNet task) provided one restricts oneself to the regime where the dataset size is much larger than the model size? Or are they specific to LLMs?
- I am conflicted by the terminology of "scaling law crossover" when the authors merely compare two protocols when scaling up compute. Scaling laws, to the best of my awareness, most often refer to power-law decay of the test loss with the considered metric. I do not understand whether the authors claim the curves are such power laws in Figs. 12-14 (the y axis is not in log scale), or whether the denomination refers to something else. Further, it is unclear whether it refers to two curves crossing, or one curve changing decay rate as compute is increased. The latter type of crossover was analyzed in simple theoretical models e.g. in (Bordelon et al, how feature learning improve neural scaling laws, 2024) in terms of training steps, and (Cui et al, Generalization error rates in kernel regression: The crossover from the noiseless to noisy regime, 2021) in terms of dataset size, which are closely related to the theoretical works discussed in the paragraph above 6.1.2.

I have very limited familiarity with the practical strategies employed in the training of large transformer models, and therefore there might be some important experimental protocol points that have eluded me. Please take my evaluation with reserve.

**Requested Changes:**

I would appreciate if the authors could address the concerns I listed above. Overall, my main concern is that some points are only illustrated by sometimes only one experimental setup, limiting their scope. Additional illustrations would allow to more convincingly establish whether the observed differences are indeed due to the regime, or are tributary to the specific tasks considered.

---

### Review · Reviewer_jCEx · 2025-08-21

**Summary Of Contributions:**

This paper discusses the shift in guiding principles for training deep learning models between the pre-LLM era and the current LLM setting. In the pre-LLM setting, overparametrization meant that hyperparameter choices were primarily aimed at reducing overfitting (i.e., reducing the generalization gap). In contrast, in the LLM era, with internet-scale datasets making models effectively underparametrized, hyperparameter choices are guided by reducing approximation error and optimizing performance scaling laws.

The paper illustrates these points with simple experiments in image classification and language model pre-training:
 - In image classification, explicit (L2) or implicit (large learning rates and small batch sizes) regularization improves performance, but this does not hold in LLM pre-training.
 - In LLM pre-training, scaling rules for optimal hyperparameter transfer do not necessarily extrapolate smoothly across scales; crossovers frequently occur where one rule begins to outperform another.
Based on these observations, the paper argues that it is unlikely that universally optimal scaling rules can be found in practice.

*Strengths*
 - The paper discusses a broad and timely topic on evolving practices in deep learning.
 - The use of simple experiments to highlight and clarify conceptual points is valuable for the community.

*Weaknesses*
 - The phenomena discussed are largely known within the ML community, so the work has limited novelty.
 - The experiments serve as illustrative demonstrations rather than as new empirical discoveries.

**Audience:**

Yes

**Audience Explanation:**

Although the paper does not identify new phenomena, its coherent discussion of training practices may still be interesting for some readers.

**Broader Impact Concerns:**

This work discusses training hyperparameters and does not have direct ethical implications.

**Claims And Evidence:**

Yes

**Claims Explanation:**

The claims are supported by experimental evidence, but most of this evidence reinforces known results.

**Requested Changes:**

To strengthen the work, I recommend expanding the language model experiments. Specifically:
 - From Fig. 11 onward, adding error bars across different initialization seeds on the evaluation loss would clarify the statistical significance of the observed differences.
 - Similarly, the extrapolation in Fig. 15 should incorporate these error bars, showing whether they account for the deviation in the observed cross-over point.

Moreover, I suggest positioning the paper more explicitly as a perspective piece, making clear that the novelty lies in framing rather than empirical discovery.

---

### Review · Reviewer_82Lt · 2025-09-06

**Summary Of Contributions:**

This paper argues that the prevailing focus of machine learning has shifted from mitigating overfitting through regularisation (“generalisation-centric”) to reducing approximation error by scaling models and data (“scaling-centric”). The authors look at common regularisation practices (L2, learning rate, batch size) and find they do not consistently improve large-scale language model pretraining. They further highlight a “scaling law crossover” phenomenon, where methods effective at small scale fail at larger scale, and discuss implications for guiding principles and model comparison in the scaling era.

Strengths
1. The problem of learning dynamics across scales is very important
2. The paper is generally well written and easy to follow

Weaknesses
1. A key weakness is the lack of definition of and clarity around “scaling”. The authors use it interchangeably to mean either relative scaling (where data >> model capacity) or absolute scaling (FLOPS). At many points it seems they are talking about absolute scale, e.g. entering a new paradigm and using FLOPs as a metric, yet also acknowledging a limitation that model capacity might catch up (relative) and a generalisation centric regime might be just beyond the horizon. This needs much better clarification.
2. The second weakness is the lack of control for confounding variables. The two compared regimes vary significantly in relative scale, absolute scale, data re-use (multiple epochs) data type (images vs natural language) architecture (CNN vs transformers) and objective (auto-regressive vs classification)
3. The presentation of the “generalization-centric” feels oversimplified. The last few years of computer vision advances were certainly heavily influenced by regularisation methods but before that, it was model scale that drove performance gains. From LeNet’s 60k parameters to AlexNet (60M) and VGG (138M), model scale grew by three orders of magnitude, far outpacing dataset size (~1M samples). Only at that point did regularisation methods become critical. So one wonders if this is a completely new regime or if we are “going back” to data >> model capacity

**Audience:**

Yes

**Audience Explanation:**

The paper topic is highly suitable for the TMLR audience, however the experiments in their current form and misalignment with claims will fall short of expectations.

**Broader Impact Concerns:**

None identified.

**Claims And Evidence:**

No

**Claims Explanation:**

1. The authors claim that regularisation techniques from generalisation centric training do not hold at different scales (it’s not completely clear if they refer to absolute scale or relative model/data scale)
2. The argument using CIFAR-10 and C4 to compare different scales is not convincing. Notwithstanding the different objectives, model architectures and data types, the x-axes are misleading. For Cifar 10 the x-axis reaches 50 epochs, with test train bifurcation around epoch 5. In the C4 example we see 100k training steps, this is around 1% of the first epoch. This means the image model sees heavy data re-use while the language model sees 0 data re-use. If we were to limit the Cifar-10 training image to the same x-scale as the NLP example (1% of first epoch), we would see two much more similar pictures, both essentially limited to the “skydiving regime”
3. The later experiments which compare image classification tasks with NLP also suffer from similar limitations. It is not clear at all whether these different results are due to relative or absolute scale differences, data differences, data re-use, architectures or objective functions. The conclusion that this is due to scale is not clear.
4. Figure 9 vs 10: It is not clear if we see the same U-shape in figure 9 if the range is extended to include the minimum, as in figure 10. Also, is that train or test error in figure 10?
5. Figure 12 is not very convincing. This result seems baked in given the choice of 2/1024, when D ~ 1024 the results will be similar and dissimilar when comparing very different learning rates. The conclusion from this experiment seems to be that “If you pick constant LR by anchoring at D=1024, it underperforms relative to 2/D as D grows.”
6. To fundamentally find the effects of scale, we need controlled experiments where scale is the variable of interest. The authors look at this in section 5, but the experiments are not set up to show this.
7. In 5.1 we see that QK-norm mitigates instability not true scale paradigms.
8. In 5.2 (see previous point), we see learning rate should scale with model size – this is partly predicted by uP theory (Yang et al).
9. In 5.3 Red varies LR as well as WD, this makes it hard to isolate the effects of WD from that of LR in the results. Proposal Blue is intended to represent µP (constant LR, constant WD), but the authors don’t 9.  demonstrate µP transferability, it’s more involved than just keeping LR and WD constant so proposal Red is not µP at all. Therefore, claiming that µP ‘fails’ at scale based on this comparison is misleading.
10.  In 5.4 there are five different variables introduced (activation function, MLP dimension, gradient norm, WD and scale), isolating the effects to one of these is impossible with the given experiment so the conclusion seems indicative at best

Taken together, the results don’t provide convincing evidence for the claims made. The experiments needs to be more carefully designed to isolate the effects of scale alone, and again the authors need to define whether they are discussing absolute scale (e.g. FLOPs) or relative scale.

**Requested Changes:**

More critical:
1. Clarify whether “scale” refers to relative scale (data vs model capacity) or absolute scale (FLOPs, parameter count), and use the term consistently.
2.	Provide controlled experiments where model scale is the only variable, or clearly acknowledge that current cross-domain comparisons (CV vs NLP) confound data type, architecture, objective, and data re-use.
3.	Improve the historical narrative by acknowledging that in CV, model scaling (LeNet → AlexNet → VGG) initially drove performance before regularisation became critical, so the “new regime” may resemble an earlier phase rather than a novel paradigm.
4.	Address the mismatch in Figure 4, where CIFAR-10 models are trained for 50 epochs while C4 models have not reached 1% of an epoch, by either normalising axes to comparable fractions of an epoch or explicitly noting this difference.
5.	In Section 5.2, clarify that the observed crossover is partly an artefact of anchoring the constant learning rate at D=1024.
6.	In Section 5.3, disentangle the effects of learning rate and weight decay scaling, and acknowledge that µP prescribes constant LR and constant WD under its parameterisation, the current comparison overstates µP’s failure.
7.	In Section 5.4, avoid bundling multiple interventions (activation function, MLP size, gradient normalisation, WD) into one experiment if the aim is to attribute effects to scale.

Less critical:

8.	Clarify figure details, including whether Figures 9 and 10 show training or evaluation loss, and extend the range in Figure 9 if needed to reveal the full U-shape.
9.	Consider rephrasing the claims/conclusions around “new paradigm” and “regularisation no longer relevant,” since the evidence may be temporary, dataset-specific, or confounded.
10.	When discussing learning rate and weight decay scaling, provide context on existing theory such as µP and µ-transfer, and frame the contributions as extensions or refinements rather than overturning prior results, unless using exact µ-transfer.

---

### Note · Authors · 2025-09-22

**Comment:**

We sincerely thank the reviewers for their valuable feedback and constructive criticism, which we will carefully consider in revising the paper. After thoughtful deliberation, we have decided to withdraw the submission in order to allow more time for improvement.

**Withdrawal Confirmation:**

I have read and agree with the venue's withdrawal policy on behalf of myself and my co-authors.